# Implementing an On-Slide Molecular Classification of Gastric Cancer: A Tissue Microarray Study

**DOI:** 10.3390/cancers16010055

**Published:** 2023-12-21

**Authors:** Simona Costache, Rebecca de Havilland, Sofia Diaz McLynn, Maria Sajin, Adelina Baltan, Sarah Wedden, Corrado D’Arrigo

**Affiliations:** 1Pathology Department, University of Medicine and Pharmacy “Carol Davila”, 020021 Bucharest, Romania; maria_sajin@yahoo.com (M.S.); adelina_baltan@yahoo.com (A.B.); 2Poundbury Cancer Institute, Dorchester DT1 3BJ, UK; rebecca@poundburycancerinstitute.org (R.d.H.); sofia@poundburycancerinstitute.org (S.D.M.); corrado.darrigo@nhs.net (C.D.); 3Pathology Department, University Emergency Hospital, 050098 Bucharest, Romania; 4Cancer Diagnostic Quality Assurance Services (CADQAS), Dorchester DT1 3BJ, UK; sarah@cadqas.org

**Keywords:** gastric cancer, molecular classification, EBER, MMR, E-cadherin, beta-catenin, p53, Her2, PD-L1, Claudin18.2

## Abstract

**Simple Summary:**

Personalised cancer treatment improves outcome for patients but requires that each individual tumour is classified using molecular tools. Extensive genomic studies for each individual patients are out of reach for the majority of the population, therefore, pathologists need to apply readily available tests to achieve molecular classification for all cancer patients. This paper describes how gastric cancer can be classified using on slide tests already used by histopathologists. Using this classification, oncologists can select more effective treatment for each patient.

**Abstract:**

*Background and Objectives:* Gastric cancer (GC) is one of the most commonly diagnosed cancers and the fourth cause of cancer death worldwide. Personalised treatment improves GC outcomes. A molecular classification is needed to choose the appropriate therapy. A classification that uses on-slide biomarkers and formalin-fixed and paraffin-embedded (FFPE) tissue is preferable to comprehensive genomic analysis. In 2016, Setia and colleagues proposed an on-slide classification; however, this is not in widespread use. We propose a modification of this classification that has six subgroups: GC associated with Epstein–Barr virus (GC EBV+), GC with mismatch-repair deficiency (GC dMMR), GC with epithelial–mesenchymal transformation (GC EMT), GC with chromosomal instability (GC CIN), CG that is genomically stable (GC GS) and GC not otherwise specified (GC NOS). This classification also has a provision for biomarkers for current or emerging targeted therapies (Her2, PD-L1 and Claudin18.2). Here, we assess the implementation and feasibility of this inclusive working classification. *Materials and Methods*: We constructed a tissue microarray library from a cohort of 79 resection cases from FFPE tissue archives. We used a restricted panel of on-slide markers (EBER, MMR, E-cadherin, beta-catenin and p53), defined their interpretation algorithms and assigned each case to a specific molecular subtype. *Results*: GC EBV(+) cases were 6%, GC dMMR cases were 20%, GC EMT cases were 14%, GC CIN cases were 23%, GC GS cases were 29%, and GC NOS cases were 8%. *Conclusions*: This working classification uses markers that are widely available in histopathology and are easy to interpret. A diagnostic subgroup is obtained for 92% of the cases. The proportion of cases in each subgroup is in keeping with other published series. Widescale implementation appears feasible. A study using endoscopic biopsies is warranted.

## 1. Introduction

There are unmet needs for a large proportion of gastric cancer (GC) patients. Classification of GC into actionable diagnostic categories would result in more effective treatment [1]. NGS-based molecular classifications, such as The Cancer Genome Atlas Program (TCGA) [2] and the Asian Cancer Research Group (ACRG) [3,4], have identified a number of important subtypes of GC characterised by different molecular signatures, including a hypermutated phenotype associated with mismatch-repair deficiency (dMMR), an ultramutated phenotype with chromosomal instability (CIN) and a genomically stable (GS) phenotype. In addition, it is apparent that there are a small proportion of cases that have integrated EBV. Nevertheless, these multi-omics classifications are difficult to implement in the diagnostic routine, and several groups have tried to identify these subtypes using a small number of on-slide tests.

The Boston group (Setia et al.) was the first to propose a hierarchical classification using a small number of on-slide tests that would stratify these patients into molecular-based and clinically actionable categories [5]. In the following years, the Boston group and others have published a number of re-iterations of this approach using minor variations in the biomarkers used and the nomenclature of the subtypes of GC identified [6,7,8]. All this work demonstrates the strength of the hierarchical approach and shows a significant degree of concordance in the subtypes and biomarkers used, but no single widely adopted classification has emerged. More importantly, there has been little attention paid to the distinction between diffuse and intestinal subtypes identified by Laurén, which remains a cornerstone of treatment decision-making [9]. For this reason, we proposed minor modifications [10], including harmonisation of the nomenclature and biomarkers used, the introduction of an indeterminate category, the addition of the predictive oncology biomarkers currently required for therapy selection and provision for new biomarkers that inevitably will become mandatory [11]. The inclusion of an indeterminate category was part of the successful implementation of the new molecular classification for endometrial carcinoma [12,13,14], and we believe it is an important element for the effective clinical implementation.

The proposed classification has six subtypes: GC associated with EBV (GC EBV+), GC with mismatch-repair deficiency (GC dMMR), GC with epithelial-to-mesenchymal transformation (GC EMT), GC with chromosomal instability (GC CIN), GC that is genetically stable (GC GS) and GC not otherwise specified (GC NOS). The proposed classification uses four groups of biomarkers: EBER ISH; the four MMR enzymes (MLH1, PMS2, MSH2 and MSH6); E-cadherin; and β-catenin and p53. In addition, the proposed classification uses a hierarchical approach for the attribution to the specific subtypes. Having identified and removed all tumours with EBV, dMMR and EMT, the use of p53 is directed to separating CIN from GS tumours. Since other markers can be used for this purpose (for instance p21) [7], the use of the nomenclature CIN and GS is more appropriate since it provides harmonisation with the work of other groups in GC and in other tumour sites [5,6,7,8]. This classification represents an inexpensive and effective tool that histopathologists can use to provide prognostic information and help in the identification of personalised cancer treatment. It is likely that histopathologists will need to deliver numerous additional predictive biomarkers in the near future. This approach will give some guidance for the prioritisation of the required companion diagnostic tests.

In order to assess the strengths and weaknesses of our proposed classification, we need to understand the feasibility of delivering these tests in a diagnostic histopathology laboratory and describe the interpretation of staining for each biomarker used and the interpretative hierarchical algorithm. For this, we used a cohort of gastric cancer resection cases in a retrospective study. We constructed a tissue microarray (TMA) library from formalin-fixed and paraffin-embedded (FFPE) tissue and evaluated a number of on-slide tests in order to classify each tumour into a defined molecular subtype.

## 2. Materials and Methods

### 2.1. Patient and Tissue Samples

We identified a cohort of 130 consecutive patients from a single hospital (University Emergency Hospital Bucharest, Bucharest, Romania) who underwent partial or complete gastrectomy for primary GC between January 2013 and February 2020, using the original histopathology report as the data source. Each case was annotated with demographic data (age and gender), histologic subtype (according to the WHO classification 5th edition, 2019 and the Laurén type, see below), grade of differentiation, stage, presence of lymphovascular and perineurial invasion and lymph node status [9,15]. All glass slides were reviewed and tumours staged using the latest edition (version eight) of The American Joint Committee on Cancer (AJCC) [16]. We could not obtain suitable well-preserved FFPE tissue or there was insufficient carcinoma left in the blocks for 51 cases; thus, a total of 79 patients were included in the final study.

This study was conducted according to the World Medical Association Declaration of Helsinki. All patients provided informed written consent for the use of their tissue in the research. The local Bioethics Committee from University Emergency Hospital Bucharest (Bucharest, Romania) approved this study (study number 31673/01.07.2020).

### 2.2. Slide Digitisation and Viewing

Glass slides were digitised using a 3D-Histech P1000 digital slide scanner (3D-Histech, Budapest, Hungary) with ×36 primary magnification and SlideCentre v3.1 (3D-Histech, Hungary). Digital slides were examined using CaseViewer v2.6 (3D-Histech, Hungary) on monitors with 4K resolution.

### 2.3. Preparation of TMA

All haematoxylin and eosin (H&E) stained tumour sections from each case were reviewed by one pathologist, and a representative slide was selected for each tumour. The selected H&E slides were then digitised and annotated digitally to identify two representative areas from each slide. TMAs were prepared using a TMA-Grandmaster (3D-Histech, Hungary) with digital recognition of the annotated areas and automatic matching of the corresponding FFPE tissue block. The TMA templates had 32 tumour cores; for orientation, we used two reference cores of liver tissue and an empty space. Each tumour was sampled in duplicate. Cores of tissue that were 1.5 mm in diameter were drilled out of the donor FFPE blocks, and a total of five TMA blocks were obtained. The H&E-stained sections from the TMA blocks were then compared with the original H&E sections to validate the TMA library.

### 2.4. On-Slide Biomarkers

FFPE sections from each TMA were cut at 3 µm and transferred onto coated slides (TOMO, Matsunami, Osaka, Japan) together with the relevant on-slide controls. They were stained using validated protocols in automated stainers. See Table 1 for a list of the reagents and the protocols of the biomarkers used. For Her2 IHC, the UltraView DAB detection system was used.

### 2.5. Evaluation of ISH and IHC Staining

Digital pathology (DP) was used to assess all the stained TMA sections (H&E and all biomarkers). The TMAs contained duplicate cores for each case. The cases were assessed as a whole, and where appropriate, the findings in duplicate cores were averaged. Where one of the cores was not assessable, only one core was used; when neither of the cores were assessable, the test was scored as “indeterminate”. The IHC and ISH-stained TMA cores were scored independently by two pathologists. Discrepancies were reviewed and discussed, and a consensus was reached. All biomarkers were assessed on invasive carcinoma tissue; in situ carcinoma was excluded from the assessment.

EBER positivity was assessed by the presence of nuclear staining in the tumour cells (TCs) [17]; the absence of chromogen in the nucleus of the TCs was defined as negative staining. Cytoplasmic staining, if present at all, was not taken into consideration.

Nuclear staining in the TCs was used to assess MMR status. Diffuse absence of nuclear staining in the TCs in the presence of staining in the internal controls (stromal cells, normal glands and inflammatory cells) was regarded as evidence of MMR deficiency [18].

E-cad and β-cat were assessed jointly. Strong circumferential membrane staining in the TCs was considered normal expression. Weak, discontinuous or absent membrane staining as well as granular cytoplasmic staining or nuclear staining in at least one of these markers were regarded as aberrant expression [19,20].

P53 was assessed in the TCs according to Köbel et al. [21]. Briefly, p53 mutation (p53m) was attributed to three abnormal staining patterns: the presence of strong and diffuse nuclear staining in at least 80% of TCs (overexpression), diffuse cytoplasmic staining with or without nuclear staining and diffuse lack of nuclear staining (complete absence). Conversely, p53 normal/wild type (p53wt) expression was characterised by nuclear staining of variable intensity admixed with TCs lacking nuclear staining.

Her2 was interpreted in line with Hofmann et al. [22] and Rüschoff et al. [23] and refers to the presence of *Her2* gene amplification. This algorithm has been adopted by the College of American Pathology (CAP) and Food and Drug Administration (FDA) [23]. The magnification rule was used to assess the intensity of membrane staining [24]. Accordingly, a positive result (3+ IHC score) was given if circumferential, basolateral or lateral membrane staining of strong intensity was discernable in ≥5 clustered TCs. An equivocal result (2+ IHC score) was given if the tumour cells showed circumferential, basolateral or lateral membrane staining of weak to moderate intensity in ≥5 clustered TCs. A negative result (0/1+ IHC score) was given for tumours with no staining or very weak segmental/granular membrane staining, irrespective of the number of TCs stained.

PD-L1 was evaluated using the combined positive score (CPS) as described in the 22C3 Agilent/Dako pharmDx interpretation guidelines and in the work of Kulangara and colleagues [25]. In CPS, both TCs and immune cells (ICs) are scored. For the TCs, only membrane staining of any intensity is included, while for ICs, both membrane and cytoplasmic staining are included. Arbitrarily, a threshold for positivity of CPS = 5 was used; a carcinoma was regarded as positive for PD-L1 if CPS ≥ 5 and negative if CPS < 5.

Claudin18.2 (CLDN18.2) was assessed by determining the proportion of TCs that showed any amount (partial or complete) of membrane staining of strong or moderate intensity (determined using the magnification rule).

### 2.6. Identification of the Molecular Subtypes Using a Hierarchical Approach

Tumours were given one of six diagnostic categories using a hierarchical approach similar to that used by the WHO classification for endometrial carcinoma [14] and by others who devised on-slide molecular classifications for GC [5,6,7,8]. This is described in detail in Costache et al. [10] and summarised in Figure 1. First, the EBER status was considered, and if positive, the case was classified as GC EBV(+); all GC EBER negative cases were then evaluated for MMR status and, if MMR-deficient, were classified as GC dMMR. All GC MMR proficient (pMMR) and EBER(−) cases were then screened for E-cad and ß-cat expression, and if this was aberrant, they were classified as GC with epithelial–mesenchymal transformation (EMT). Following this, all tumours with preserved E-cad and ß-cat expression, pMMR and EBER(−) were assessed for p53 status; those showing a p53m pattern were classified as GC CIN, while those with a wild-type staining pattern were classified as GC GS.

With this hierarchical approach, once a tumour is assigned to a molecular subgroup, the downstream biomarkers are non-contributory for classification purposes. Only when a downstream biomarker of a tumour not yet classified cannot be interpreted (indeterminate result), the case is placed in the indeterminate category (GC NOS).

### 2.7. Assessment of Laurén Type

The Laurén type was determined on the whole sections and on the TMAs using the two-tier system (intestinal and diffuse) according to Laurén [9]. The third category of “unclassified” from Laurén was later divided by Carneiro and colleagues into solid type and mixed type [26]. We classified the solid tumours as intestinal and the mixed tumours according to the majority component. Mucinous (colloid) carcinoma was included in the intestinal type if the formation of glands or trabeculae was present or in the diffuse type if only a single-cell pattern was present [9,26].

## 3. Results

### 3.1. Cohort Description

In our cohort of 79 patients, 66% were male and aged between 42 and 83 years (median age 65.5 years), and 34% were females and aged between 47 and 83 years (median age 67 years). The predominant Laurén subtype was intestinal (75%). The WHO classification [15] recognises five main histological subtypes of gastric adenocarcinoma: tubular, papillary, poorly cohesive (including signet ring), mucinous and mixed. In our series, the most common histological subtypes were tubular and papillary (together 44%), followed by poorly cohesive carcinoma (25%), mixed carcinoma (20%) and mucinous (7.5%). Using the two-tier system for tumour grading [15], 75% of our cohort had high grade (poorly differentiated), and 25% had low grade (moderate and well-differentiated).

Regarding staging, most tumours were pT3, invading the subserosal space (60%), followed by pT4 (34%), pT2 (5%) and pT1 (1%). The most common regional lymph node stages were pN2 and pN3 (each 31%), followed by pN0 (19%) and pN1 (17%). The greatest majority of cases (94%) showed carcinoma within vessels (lymphatics and/or veins), and 84% of cases had perineural infiltration.

### 3.2. Molecular Classification

There were 5 cases (6%) of GC EBV(+), 16 cases (20%) of GC dMMR, 11 cases (14%) of GC EMT, 18 cases (23%) of GC CIN, 23 cases (29%) of GC GS and 6 cases (8%) of GC NOS. Overall, 10 cases had at least one indeterminate test, but only six cases fell into the GC NOS category, as the classification of the other four cases was unaffected. All indeterminate tests (and by extension, diagnoses of GC NOS) were due either to TMA block failure at microtomy or loss of tissue during staining. Laurén intestinal type was prevalent (≥80%) amongst all molecular subtypes except for the GC EMT (9%). These results are summarised in Table 2.

### 3.3. Companion Diagnostic Predictive Biomarkers

We assessed three predictive biomarkers: Her2, PD-L1 and claudin 18.2. Examples of positive and negative cases are in Figure 2.

The Her2 status could not be determined in six cases (8%) due to TMA failure. There were four cases (5%) that were Her2 positive on IHC (3+), and four cases (5%) that were Her2 equivocal on IHC (2+); DDISH studies on these four equivocal cases showed that all had *Her2* gene amplification with numerous large amplification clusters throughout and, therefore, were classified as Her2 positive. All the Her2 positive cases were of Laurén intestinal type. They were distributed amongst GC dMMR (1 case), GC EMT (1 case), GC CIN (3 cases) and GC GS (3 cases), and there were none amongst GC EBV(+) and GC NOS.

The PD-L1 status could not be determined in six cases (8%) due to TMA failure; these were the same cases where the Her2 status could not be determined. We found 16 cases (22%) positive for PD-L1 (CPS ≥ 5) with the highest prevalence amongst GC EBV(+) (40%) and GC dMMR (33%), followed by GC GS (21%), GC EMT (20%) and GC CIN (17%).

The CLDN18.2 status could not be determined in seven cases (9%) due to TMA failure or an insufficient number of tumour cells; these included all the cases where Her2 and PD-L1 status was indeterminate. There were 41 cases (57%) with CLDN18.2 in <40% of the TCs, 11 cases (15%) with Claudin 18.2 in ≥40% of the TCs but <75% of the TCs, and 20 cases (28%) with CLDN18.2 in ≥75% of the TCs. Overall, there were 31 cases (43%) with expression of CLDN18.2 in at least 40% of the TCs; the highest prevalence of cases with ≥40% TCs was in GC GS (77%), followed by GC EBV(+) (60%) and GC CIN (44%), while the prevalence amongst GC dMMR (33%) and GC EMT (20%) appeared lower. Importantly, the prevalence amongst the Laurén intestinal type of tumours with ≥40% TC expression of CLDN18.2 was 48%, while that of the Laurén diffuse type was 30%. These findings are summarised in Figure 3.

## 4. Discussion

We set out to understand the feasibility of implementing the improved on-slide classification for GC that we proposed [10]. In particular, we wanted to define the interpretative algorithms for each of the biomarkers used and assess the challenges in interpretation. For this, we used tissue from a cohort of 79 cases of GC resection, and we constructed TMAs. In addition, we wanted to determine the relative proportion of cases in each of the subcategories and their association with the Laurén subtypes and with the predictive biomarkers currently used for the management of GC patients.

Our results demonstrate that, in the majority of cases (92%), GC can be classified into one of five molecular subtypes. We did not find GC EBV(+) that were also dMMR, and this is consistent with other published data [27,28]. The multi-omics studies [2,3,4] provide evidence for the hierarchical approach; in fact, tumours with multiple defects retain a molecular signature typical for a specific subtype. This hierarchical approach is necessary in our classification, since a proportion of GC dMMR is also p53m (38%), and some dMMR cases (12%) also have aberrant expression of E-cad or β-cat. Furthermore, this hierarchical approach proved useful in cases with test failure; in fact, we were still able to reach a specific diagnostic category in 40% of such cases because the specific test failure was over-ruled by a hierarchically more important biomarker that did not fail. GC NOS/indeterminate represents 8% of our total; this was due to test failure for MLH1 (two cases), E-cad (three cases) and β-cat (one case).

In our cohort, GC EBV(+) has a prevalence of 6%. This compares favourably with Setia et al. (2016) who found EBV(+) in 5% of their cases with the TCGA (9%) and with Ramos et al. (10%) [2,5,8]. To determine the EBV status, we chose EBER ISH. In the context of a classification for widespread adoption, the use of IHC may be more desirable; however, the sensitivity (44%) and specificity (93%) of IHC for the latent membrane protein 1 (LMP1) compare unfavourably with EBER ISH (sensitivity 94% and specificity of 69%) [29]. Other techniques, such as sequencing, microRNA and droplet digital PCR, may be more specific, but EBER ISH is considered the gold standard for detecting and localising latent EBV in FFPE tissue [30,31,32]. The role of EBV in GC remains uncertain, and it is possible that it has an involvement in a larger number of GC cases through a “hit and run” mechanism [33]. What has been clear from the comprehensive molecular assessments of the TCGA is that, in a smaller group of GC patients, the tumour retains EBV-related molecular pathways, which correlate with specific outcomes and responses to treatment [34,35]. For these reasons, the use of EBER ISH is appropriate to recognise these cases.

Systematic sequencing in different tumour sites (for instance endometrium and colorectum) has shown the presence of independent molecular pathways with broad commonalities, such as chromosomal instability (CIN), CpG island methylator phenotype (CIMP) and defects of double-stranded DNA repair (MMR deficiency) [36]. In addition, a smaller proportion of cases seem to have oncogenic drivers specific to the tumour site, for instance, Her2 amplification in breast cancer or POL-E mutation in endometrial carcinoma [37,38]. In this context, GC has a small number of cases where EBV appears to be the oncogenic driver; these tumours have extremely high CIMP and are molecularly, genetically and epigenetically distinct from all other types [39]. Unlike the ACRG study, the TCGA group recognised this as a unique category with a good prognosis, distinct pathological features and an association with a good response to immune checkpoint therapy [40,41]. We believe that GC EBV(+) should be recognised as a distinct category as the first step in the hierarchical approach.

The proportion of dMMR tumours in our cohort is 20%. Similar proportions were found in other studies [2,3,5,6,7,8] with the prevalence ranging between 16% (Setia et al.) and 24% (Ahn et al.). In our study, the predominant cause of dMMR was loss of MLH1 (94%), followed by MSH2 (6%). This is concordant with published data, which identify hypermethylation of the MLH1 promoter as the most common mechanism of dMMR in GC with MSH2 mutation being present in only a minority of cases [42].

Absence of IHC staining for the MMR enzymes MLH1, PMS2, MSH2 and MSH6 is widely accepted as a surrogate marker of mismatch-repair status and correlates well with previously used surrogate markers, such as microsatellite instability and sequencing [18]. In our diagnostic routine, we assess MMR status using all four IHC markers; however, for this work, we used only two of the biomarkers (MLH1 and MSH2), principally in order to save tissue sections. Since MLH1 promoter methylation and germline mutation in MLH1 and MSH2 are the most frequent causes of MMR deficiency in gastric cancer, some institutes use only MLH1 and MSH2 in the diagnostic routine [43]. In our experience, the use of all four markers aids interpretation, and we would not advocate implementing GC classification using only two of these biomarkers. For GC dMMR, it may be desirable to perform MLH1 promoter methylation studies to help distinguish syndromic patients from sporadic cases; this distinction is not part of our molecular classification. It may be worth noting that, within the GC dMMR group, we identified two cases (12%) that also showed aberrant expression of E-cad or β-cat; similar cases were also described in the cohort assessed by Setia et al. [5]; the hierarchical approach ensures that such cases can be firmly attributed to a specific category, in this case GC dMMR.

In this study, GC EMT represents 14% of the cohort. This category comprises all cases with aberrant expression of E-cad and/or β-cat. The majority of these cases (10/11 or 91%) are of Laurén diffuse type. The prevalence of GC EMT is slightly higher than that found by Ramos et al. (9%) [8] but similar to that found in the ACRG study (15%) [3,4] and in the study of Ahn et al. (15%) [6]. However, it is much lower than the incidence found in other studies, which ranged between 20% and 29% [2,5,7]. One case in our GC EMT cohort was of Laurén intestinal type. On review, this case showed tubular morphology. There are reports of tubular carcinoma of the stomach with aberrant E-cad expression [6,7,8]. It is difficult to argue that tubular carcinoma shows morphological evidence of EMT, since it has well-developed epithelial structures; nevertheless, in comprehensive genomic studies, it has gene expression profiles more similar to tumours with more classic EMT [3,4]. This may signify that GC EMT may not be the most appropriate term to identify this group of tumours.

In order to identify cases of GC EMT, we used β-cat in addition to E-cad. Others, who attempted molecular classification of GC with on-slide biomarkers, limited testing to E-cad only [5,6,7,8]. EMT is linked to loss of cell-to-cell adhesion, and this is, in most cases, due to a defect in E-cad. A prototype tumour with EMT is lobular carcinoma of the breast (LBC), which can be associated with sporadic or familiar defects in *CDH1*, the gene encoding for E-cad. Nevertheless, in a small proportion (10–15%) of LBC, the defect lies in accessory molecules of the cadherin–catenin complex, which include α-catenin, β-catenin and p120 catenin [44]. The addition of IHC for β-cat has proven helpful in understanding the loss of cell-to-cell adhesion [45,46], and both markers are now used routinely in breast pathology, providing a more accurate diagnosis of LBC [47,48].

There are strong similarities between LBC and diffuse GC. A proportion of diffuse GC is hereditary, part of a rare autosomal dominant syndrome that was first described in 1998, and is characterised by an increased risk of diffuse GC and LBC [49]. The most common underlying defect in this syndrome is mutation in the *CDH1* gene, followed by mutation in the *CTNNA1* gene that encodes for α-catenin [50]. Alpha catenin defects result in destabilisation of the cadherin–catenin complex with increased degradation of these molecules and lead to abnormal localisation of β-cat [51,52]. For these reasons, we added β-cat to our panel. Others have shown that aberrant expression of β-cat is also linked with a defective cadherin–catenin complex in GC, but this marker is yet to be in routine use [53].

In our study, 64% of GC EMT (7/11) show loss of both E-cad and β-cat expression, 18% (2/11) show only aberrant E-cad expression, and in the remaining 18% (2/11), E-cad is indeterminate while β-cat shows aberrant expression. Our experience supports published evidence that the assessment of EMT is greatly facilitated by the use of both markers [54]. In addition, a promising new immunotherapy targeting CLDN18.2 appears particularly effective in diffuse GC. CLDN18.2 is a constituent of tight junctions and becomes accessible to the immune system when tight junctions are not functional, as in the case of GC EMT. Zolbetuximab, a humanised monoclonal antibody, selectively binds CLDN18.2 on tumour cells and mediates antibody-dependent, cell-mediated cytotoxicity (ADCC); this treatment is more effective in tumours with a higher expression of CLDN18.2 [55,56,57]. The correct identification of GC EMT is important not only for its association with a poorer prognosis but also for the selection of specific immune therapy.

CLDN18.2 has the potential to become an important predictive biomarker in view of its effectiveness in clinical trials [55,56,57]. There is some uncertainty over the threshold for positivity. Some trials considered a case positive for CLDN18.2 if there was moderate or strong intensity in ≥40% of the TCs; others were positive if in ≥70% of the TCs or in ≥75% of the TCs [58]. For the purpose of this analysis, we considered a threshold for positivity of ≥40%. It is possible that some of the molecular subgroups may have a better response to this new drug; nevertheless, this requires further work.

GC CIN represents 23% of our cohort, which is lower than that found by Ramos et al. (35%) and the ACRG (36%). It is worth noting that, in both studies, there is no indeterminate group. If the p53m cases of the GC NOS group are added to the GC CIN group, the total (27%) approaches that of these two studies [3,8]. In our study, the proportion of p53m cases is considerably lower than that in the work of Setia et al. (51%) and Ahn et al. (49%) [5,6].

The number of cases in the GC GS category in our study (29%) compares well with that of Ramos and colleagues (29%), Ahn et al. (21%) and the ACRG study (26%) but is very different to that reported by Setia et al. (7%) [3,5,6,8]. These differences are not surprising, since the p53 mutation is assessed with IHC in three of these studies and relies on sequencing in the ACRG study. In addition, the staining parameters and interpretative algorithms are different, and these are known to have a significant impact on results. For example, both Setia et al. and Ahn et al. did not recognise cytoplasmic staining as aberrant expression linked to mutation. We used more recent guidelines, which have been developed for p53 assessment in endometrial carcinoma [21].

Eight of our cases (10%) are Her2 positive (IHC 3+ and DDISH amplified). Other studies reported a prevalence between 3% and 5% [3,6]. Our study is small, but we found no association with any particular molecular subtypes. However, all cases were of Laurén intestinal type, confirming previous observations [5,6,7,8] and highlighting the importance of the Laurén model to determine further downstream tests.

We found a total of 16 PD-L1 positive cases in our cohort. These results should be interpreted with caution, as PD-L1 staining in tumour tissue can be heterogeneous, and smaller samples (such as those in our TMAs) may have significant sample bias. In addition, we chose a positivity threshold of CPS 5 in view of the approval of nivolumab in GC. There is a second clinical threshold for GC at CPS 10 for the use of pembrolizumab. Our study showed that GC EBV(+) is enriched with PD-L1 positive cases (40% vs. 21% in GC EBV(-)), and this is in keeping with other published data [2,5,6]. Some authors have linked this to high numbers of lymphoid cells in the stroma of EBV(+) GC [59]. In our study, the majority (81%) of PD-L1(+) GC is of the Laurén intestinal type and only three of sixteen cases (19%), signifying no preferential expression in either subgroup. There is a variation in the percentage of PD-L1 overexpression in the literature, ranging between 28 and 65% for the intestinal type and between 19 and 54% for the diffuse type. Comparison is difficult because of the different thresholds for positivity that have been used [60,61,62,63].

This study shows that a molecular classification is possible to implement using existing histopathology resources and expertise. While this study has a very small number of cases and further work using larger cohorts is needed, the correlation between molecular subgroups and prevalence of key predictive oncology biomarkers is apparent. The clinical value of such an association in the context of busy clinical units cannot be overlooked, especially when some of these predictive biomarkers are send-away tests with long turnaround times. Knowledge of the molecular subgroup and, hence, an understanding of the prognosis as well as the prevalence of specific predictive tests would greatly assist the multi-disciplinary team (MDT) discussions and aid treatment decision-making while awaiting the results of predictive tests.

There are limited data on the biological behaviour of each of the molecular subtypes of GC. An understanding of prognosis is difficult in consideration of the fact that some of these subgroups are enriched for responders to immunotherapy. The relative aggressiveness of each subgroup has been estimated from published papers [2,3,4,5,6,8], and we have attempted to collate the available data in Figure 4, which provides an indication on the aggressiveness of each subtype. This picture may change when considering the relative abundance of responders in each category and the effectiveness of existing biological therapy (for instance herceptin, pembrolizumab and nivolumab) [23,64,65] and new biological drugs (for instance zolbetuximab) [55,56,57].

The strength of this work includes the hierarchical approach that has allowed for us to classify cases even in the event of test failure, the inclusion of a second biomarker for the identification of cases with EMT that strengthens the accuracy of this diagnosis, the incorporation of traditional classifications (Laurén) that remain pertinent for clinical management, the harmonisation of the nomenclature of the molecular subtypes and the provision for the use of predictive biomarkers for therapy selection. This is the first work that integrates CLDN18.2 into the context of a molecular classification.

This study has a number of limitations. We have no outcome data for our cohort of patients; therefore, the assumption on the prognostic significance of these categories is inferred from other studies. This study uses TMAs, which may be suboptimal when assessing heterogenous tumours, and the relatively small cohort means that some subgroups have a very low number of events. The tissue used might have been affected from suboptimal pre-analytics, since these are derived from gastric resection specimens where fixation and processing were not controlled. Finally, this study was not designed to assess the accuracy of interpretation of the biomarkers used.

## 5. Conclusions

Our study demonstrates that a hierarchical molecular classification that uses five on-slide biomarkers on FFPE tissue can be performed in a histopathology laboratory, can be interpreted by histopathologists and has the potential to be used in the diagnostic routine. Our results using this classification are in line with other published work. The definition of the parameters for the interpretation of these tests, the establishment of an indeterminate category and the inclusion of predictive biomarkers for treatment selection represent an improvement over previous studies. Our hierarchical classification may represent an effective and affordable screening tool for personalised treatment of gastric cancer. We are currently undertaking a study using endoscopic biopsy to understand if this classification can be implemented using the diagnostic samples.

We believe it is important to ensure that molecular classification for GC becomes established worldwide, and to this extent, consideration should be given to create a repository of digital slides from studies such as this that could be freely available for review by other groups. This may encourage others to become familiar with this work and, hopefully, facilitate the uptake.

## Figures and Tables

**Figure 1 cancers-16-00055-f001:**
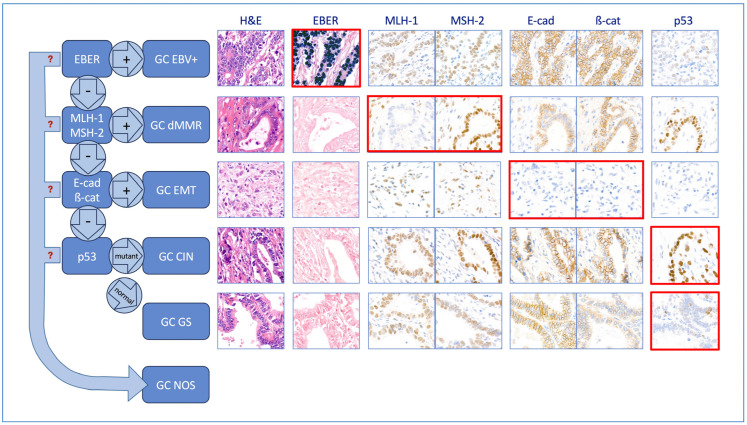
Hierarchical classification. On-slide biomarkers are assessed sequentially; the micrographs framed with a red line highlight the critical test results. Tumours are placed into the GC EBV(+) group if EBER is positive (strong and diffuse blue nuclear staining in the micrograph framed with a red line); if EBER is negative, MMR markers are assessed, and if found to be deficient, tumours are classified as dMMR (in the tiles framed with a red line, please note absence of MLH-1 nuclear staining in the tumour glands compared with the presence of staining of MSH-2); if MMR proficient, E-cadherin and β-catenin are next assessed, and if expression is aberrent, tumours are classified as GC EMT (this is exemplified in the tiles framed with a red line, which show diffuse absence of these two biomarkers as compared with the tumours in the other categories that show strong membrane expression); lastly, if EMT marker expression is normal, tumours are assessed for p53 mutation; if a mutant pattern expression is detected, they are classified as GC CIN (see strong and diffuse nuclear staining in >80% of the tumour cells in the micrograph framed with a red line); if wild-type p53 expression is detected, they are classified as GC GS (note focal nuclear staining with variable intensity in some of the tumour in the last tile framed with a red line). All microphotographs taken at ×20 magnification using a digital microscope.

**Figure 2 cancers-16-00055-f002:**
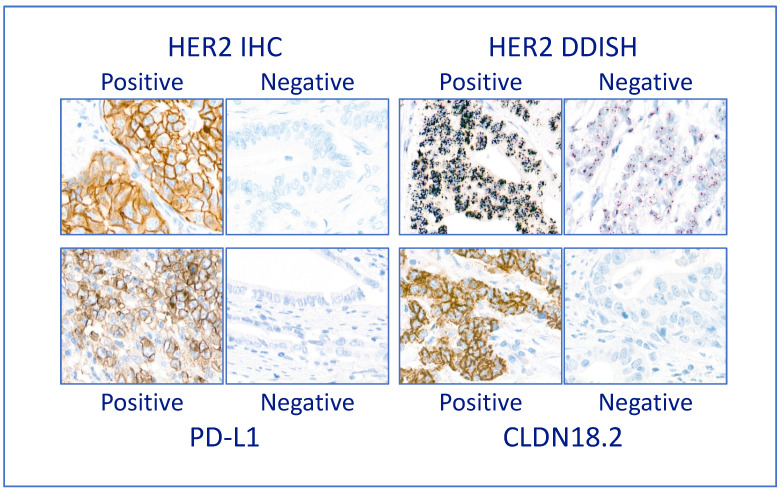
Predictive on-slide biomarkers. Examples of positive and negative cases of the three predictive biomarkers assessed, photographed at ×400 magnification. The case positive for Her2 IHC shows the presence of complete membrane staining of strong intensity in the TCs; the case positive for Her2 DDISH shows the presence of large clusters of amplification (denoted by silver black dots) in each TC; the case positive for PD-L1 shows membrane staining in the TCs as well as some staining in the ICs; the case positive for CLDN18.2 shows strong membrane staining in most TCs.

**Figure 3 cancers-16-00055-f003:**
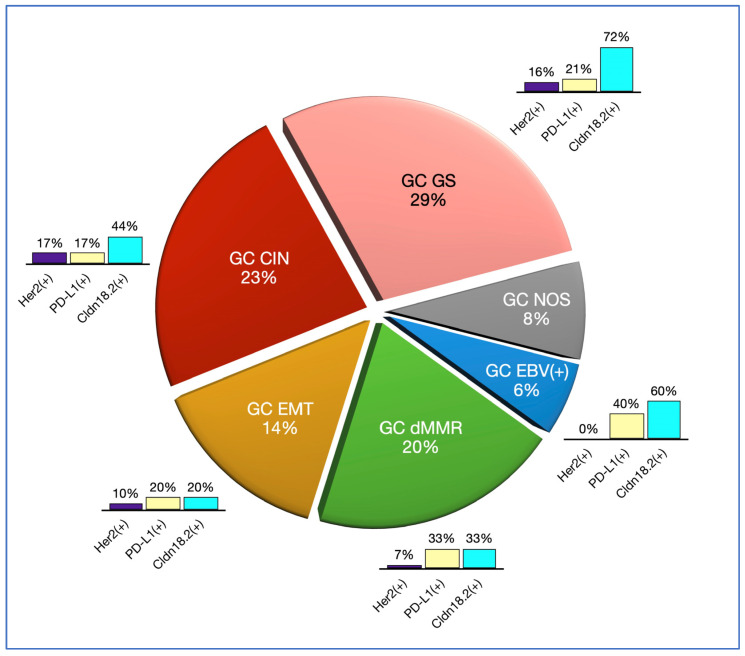
Molecular classification of our GC study cohort. The pie chart indicates the relative prevalence of each molecular subtype; the bar graphs indicate the prevalence of specific predictive biomarkers (Her2, PD-L1 and Claudin18.2—CLDN18.2) in each of the molecular subtypes.

**Figure 4 cancers-16-00055-f004:**
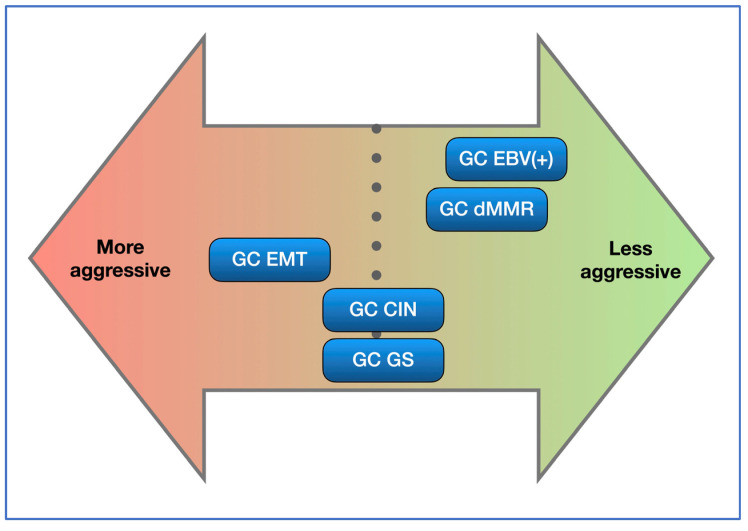
Molecular subgroups and their aggressiveness. This is an indicative diagram considering published papers [2,4,5,6,7,8]. GC EBV(+) and GC dMMR are the two less-aggressive subtypes, while GC EMT is the most-aggressive subtype. Availability of effective treatment may alter outcomes and, therefore, the prognosis.

**Table 1 cancers-16-00055-t001:** Lists of reagents used for on-slide tests and protocols.

Test	Type	Vendor, Catalogue Number	Clone	On-Slide Control	Conc. or RTU	Retrieval Solution and Time (min)	Antibody Incubation Time (min) and T°	Platform
EBER	ISH	Ventana, 800-2842	-	Cell lines (Ventana)	As per package insert			
MLH-1	IHC	Leica, PA0988	ES05	Appendix	RTU	AR1, 30	15 min RT	Bond III
MSH-2	IHC	Leica, PA0989	79H11	Appendix	RTU	AR2, 30	15 min RT	Bond III
E-cadherin	IHC	Leica, PA0387	36B5	Skin	RTU	AR2, 20	15 min RT	Bond III
Beta catenin	IHC	Leica, PA0083	17C2	Skin	RTU	AR1, 20	15 min RT	Bond III
p53	IHC	Leica, P53-DO7-L-CE	DO-7	Colon AC	1:200	AR2, 30	15 min RT	Bond III
Her2	IHC	Ventana, 790-4493	4B5	Cell lines (Histiocyte)	RTU	CC1, 36	16 min 36 °C	Benchmark Ultra
DDISH	ISH	Ventana, 800-6043	-	Cell lines (Histiocyte)	As per package insert			
PD-L1 (22C3)	IHC	Agilent, SK006	22C3	Cell lines (Histiocyte)	RTU	TRS low pH, 20	30 min RT	Dako Autostainer Link 48
Claudin18.2	IHC	Ventana, 790-7027	43-14A	Stomach	RTU	CC1, 64	16 min 36 °C	Benchmark

ISH = in situ hybridisation; IHC = immunohistochemistry; AC = adenocarcinoma; RTU = ready to use; T° = temperature; AR1 and AR2 = Leica antigen retrieval solution 1 and 2; CC1 and CC2 = Roche Ventana cell conditioning solution 1 and 2; TRS = Dako Agilent target retrieval solution low pH.

**Table 2 cancers-16-00055-t002:** Molecular classification of the study cohort.

Molecular Type	Prevalence	Laurén’s Type
GC EBV(+)	6%	Intestinal: 80%
		Diffuse: 20%
GC dMMR	20%	Intestinal: 88%
		Diffuse: 13%
GC EMT	14%	Intestinal: 9%
		Diffuse: 91%
GC CIN	23%	Intestinal: 94%
		Diffuse: 6%
GC GS	29%	Intestinal: 87%
		Diffuse: 13%
GC NOS	8%	Intestinal: 50%
		Diffuse: 50%

## Data Availability

All data are available on requested.

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
