# Peer review of "Implementing an On-Slide Molecular Classification of Gastric Cancer: A Tissue Microarray Study"

_cancers, 2023, doi:10.3390/cancers16010055_

Round 1
Reviewer 1 Report
Comments and Suggestions for Authors
The manuscript entitled "Implementing an on-slide molecular classification of gastric cancer: a tissue microarray study"
Major issues:
1. In the present study, there is nothing new disclosed. The authors just followed the original classification and performed the IHC study in a quite limited number of cases.
2. The authors should try to correlate their findings to certain important clinicopathological features, such as H.P infection, tumor stage, or the expression status of emerging biomarkers such as FGFR2b, CLDN18.2, ... etc.
3. The since some IHC markers are known to show intratumoral heterogeneity. The author should confirm the TMA is adequate to represent the result of the whole section study.
4. To get convincing results, I would suggest the authors should use CDx antibodies for the evaluation of dMMR.
Comments on the Quality of English LanguageEnglish editing is mandatory.
Reviewer 2 Report
Comments and Suggestions for Authors
The manuscript with ID cancers-2702495 presents relevant data on one of the most prominent illnesses that record an increased mortality rate, which is gastric cancer. In the present work, the authors proposed a modified classification of GCs in order to choose the most appropriate treatment, thus focusing specifically on the on-slide biomarkers assays available in histopathological determinations. The proposed classification is promising as the results suggest up to 92% subgroup diagnosis. In addition, the authors present the limitations of their study. Still, some aspects should be clarified in the present form of the manuscript: the authors should emphasize the strengths of their research along with the limitations that have been already mentioned; Figure 1 must be detailed and explained (each slide must be discussed); where any other criteria considered for the patients' selection?
Reviewer 3 Report
Comments and Suggestions for Authors
Dear Editor,
I reviewed the manuscript by Costache et.al., “Implementing an on-slide molecular classification of gastric cancer: a tissue microarray study”
This study focuses on classification gastric cancer using markers that are widely available in Histopathology and provides a new concept to easy interpret. This manuscript is well written, the study is well designed and the data are repented in best form. In addition this study will be interesting for the reader of the journal and according , can be published in the present form.
many thanks
Reviewer 4 Report
Comments and Suggestions for Authors
The authors of the original article aim to create a tissue microarray (TMA)-based study using surgical specimens of gastric cancer resections by selecting tissue cores representative of the tumor. A molecular-based hierarchical classification system is being tested on TMAs, which can be applied in practice. The classification consists of the following groups: EBV-positive, defective MMR status, EMT characteristics, CIN type, chromosomally stable, and unclassifiable gastric cancer types.
The technical implementation of the TMAs has been carefully and appropriately performed.
The IHC or ISH methods used for the identification of each group were well chosen, the design of IHC and ISH was appropriate, and the evaluation criteria were carefully set up according to international recommendations and consistently followed in the analysis of the samples. The molecular classification was compared with the Lauren classification.
The selected positive controls are also appropriate.
Histological and ISH images are of good quality.
For the 79 samples suitable for TMA analysis, molecular subgroups occurred in more or less the same number of samples as in the previous relevant working group data. For these differences, the explanation for the differences due to methodological differences is understandable.
The authors correctly conclude that multi-omics studies favor a hierarchical approach, as tumors with multiple defects retain the molecular signature of the subtype. The hierarchical approach has also proved useful for technically unevaluable samples, as in 40% of such cases, a specific diagnostic category could still be reached as the specific test success was overridden by a hierarchically more important biomarker that did not fail.
The results are discussed in a fair way, with potential failures in mind.
The limits of the study are also identified realistically, and the results are interpreted with limits in mind.
The design and execution of the study, the methods used, the results obtained, and their interpretation are fully adequate and correct.
In particular, the authors point out that the number of cases studied is relatively small. Similar to gene expression databases (e.g., GEO), do the authors of the article think that it would be worthwhile to create an international TMA database available online, in which all digitized TMAs could be uploaded and histological samples from other groups could be re-evaluated and re-analyzed? This should be included at the end of the discussion.
All in all, this is a very well-designed and well-executed study, the results of which could be incorporated into daily practice if validated in studies with larger sample sizes.
